# Facilitating Mitophagy via Pink1/Parkin2 Signaling Is Essential for the Neuroprotective Effect of β-Caryophyllene against CIR-Induced Neuronal Injury

**DOI:** 10.3390/brainsci12070868

**Published:** 2022-06-30

**Authors:** Jiangyan Rao, Yu Wu, Xiaomei Fan, Shaonan Yang, Lu Jiang, Zhi Dong, Sha Chen

**Affiliations:** The Key Laboratory of Biochemistry and Molecular Pharmacology, College of Pharmacology, Chongqing Medical University, Chongqing 400010, China; rjy1988@email.swu.edu.cn (J.R.); 2020111584@stu.cqmu.edu.cn (Y.W.); 2020111571@stu.cqmu.edu.cn (X.F.); yangshaonan@stu.cqmu.edu.cn (S.Y.); jianglu@stu.cqmu.edu.cn (L.J.); 100798@cqmu.edu.cn (Z.D.)

**Keywords:** mitophagy, β-caryophyllene, cerebral ischemia-reperfusion, Pink1/Parkin2 pathway, neuronal injury

## Abstract

Mitophagy is an important mechanism for maintaining mitochondrial homeostasis through elimination of damaged or dysfunctional mitochondria following cerebral ischemia-reperfusion (CIR) injury. β-Caryophyllene (BCP) is a natural sesquiterpene compound found in the essential oil of plants and has been shown to ameliorate CIR injury. However, whether BCP protects neurons from CIR injury by activating mitophagy is still unclear, and the underlying mechanism remains unknown. In the present study, a mouse neuron HT-22 cell of oxygen-glucose deprivation/reoxygenation (OGD/R) and C57BL/6 male mouse of transient middle artery occlusion followed by 24 h reperfusion (MCAO/R) were established the model of CIR injury. Our results show that BCP remarkably protected against cell death and apoptosis induced by OGD/R, and decreased neurologic injury, infarct volume, and the injury of neurons in CA1 region on MCAO/R mice. In addition, BCP accelerated mitophagy by regulating expression of mitochondrial autophagy marker molecules and the mt-Atp6/Rpl13 ratio (reflecting the relative number of mitochondria), and promoting autophagosome formation compared with OGD/R and MCAO/R groups both in vitro and in vivo. Furthermore, this study revealed that BCP pre-treatment could activate the Pink1/Parkin2 signaling pathway, also with mitophagy activation. To explore the mechanisms, mitochondrial division inhibitor-1 (Mdivi-1) was used to investigate the role of BCP in CIR injury. We found that Mdivi-1 not only decreased BCP-induced facilitation of mitophagy, but also significantly weakened BCP-induced protection against OGD/R and MCAO/R models, which was consistent with levels of Pink1/Parkin2 signaling pathway. Taken together, these results suggest that facilitating mitophagy via Pink1/Parkin2 signaling is essential for the neuroprotective effect of BCP against CIR injury.

## 1. Introduction

Stroke, including hemorrhagic and ischemic stroke, is a major affliction threatening human health that has become the second most common disease in the world [1]. Ischemic stroke accounts for more than 80% of stroke occurrences and is a leading cause of mortality and disability [2]. Currently, the most effective treatment is timely thrombolysis for ischemic stroke [3]. However, at most around 8% of stroke patients eligible for recombinant tissue plasminogen activator intravenous thrombolysis receive it owing to its limited time window for treatment [4]. The reperfusion stage after cerebral ischemia is prone to cause further damage by mechanisms including free radical production, oxidative stress, excitatory amino acid toxicity, and intracellular calcium overload, which promote the release of a large number of neurotransmitters, ultimately leading to neuron damage and death [5]. Therefore, compared with thrombolytic therapy, enhancing the self-resistance and protection of neurons for ischemic stroke has broader application prospects because it has no treatment-time limitation.

Mitochondria are known as the “power plants” of cells because they play a key role in maintaining cell energy homeostasis and, thus, are certainly involved in neuronal death following ischemic stroke [6]. Recent studies showed that CIR induced prominent mitochondrial dysfunction, including mitochondrial permeability transition pore opening, the damage of mitochondrial morphological, and Ca^2+^-mediated mitochondrial swelling to initiate neuronal death [7]. Moreover, mitochondrial dysfunction is closely related to many biological processes in the cell, such as apoptosis, oxidative stress, and inflammation in the occurrence and development of CIR. Furthermore, the accumulation of damaged mitochondria induces further brain damage, thus forming a vicious cycle [8]. Therefore, strict control of mitochondrial quality and quantity is crucial to avoid the pathological effects of dysfunctional mitochondria on neurons [9]. Failure to remove damaged mitochondria will lead to cell death, so an efficient clearance of damaged mitochondria is critical to mitigate CIR-induced nerve damage.

Mitochondrial autophagy, also known as mitophagy, is a biological process by which select mitochondrial cargos are wrapped by mitochondria-derived vesicles and delivered to lysosomes for degradation, an essential process for maintaining mitochondrial contents and metabolic homeostasis [10]. Mitophagy is essential for the renewal and redistribution of mitochondria, regulating the homeostasis and biogenesis of mitochondrial, controlling the number and quality mitochondrial [11]. Increasing numbers of reports have demonstrated that mitophagy is associated with neurodegenerative diseases and brain injury [12]. Elimination of damaged mitochondria efficiently by activated mitophagy seems to be critical for neuronal survival following CIR injury [13,14,15,16,17].

Cannabinoid receptor type 1(CB1R) and cannabinoid receptor type 2 (CB2R) all are cannabinoid receptors. CB2R agonists have shown against CIR by its neuroprotective effects [18]. β-caryophyllene (BCP), a natural sesquiterpene compound found in the essential oil of plants, is reportedly a CB2R-selective agonist [19]. Previous research revealed that BCP can protect neurons against CIR injury in mice by promoting polarization of microglia toward an M2 phenotype [20]. BCP can also alleviate CIR injury by activating the PI3K/Akt signaling pathway [21]. Moreover, BCP elicited a neuroprotective effect against CIR injury by regulating neuronal death and inflammation both in vivo and in vitro [22]. However, whether the protective effects of BCP on CIR injury are mediated by regulation of mitophagy to maintain mitochondrial homeostasis and biogenesis is still largely unknown.

The PTEN-induced putative kinase 1 (Pink1)-Parkin (E3 ubiquitin ligase) signaling pathway is the classical regulatory mechanism for mitophagy. Pink1 is kept at a low level depending on rapid degradation by mitochondrial proteases. In the context of mitochondrial damage, Pink1 is on the outer membrane of mitochondria and recruits Parkin from cytosolic to the outer membrane of mitochondria too. Parkin mediates engulfment and subsequent destruction of mitochondria through autophagosomes when it transferred to the mitochondrial surface. Subsequently, Parkin is recognized by p62 and bound this adaptor protein. p62 recruits ubiquitinated substances into autophagosomes by binding to LC3, ultimately leading mitochondria being degraded by lysosomes. Additionally, recent evidence shows that upregulation of Pink1/Parkin-mediated mitophagy can alleviate CIR injury [17,23,24]. However, whether the Pink1/Parkin pathway is crucial for the protective effects of BCP on CIR injury is still largely unknown. Thus, our study aimed to determine whether BCP attenuates CIR-induced neuronal injury by facilitating mitophagy via the Pink1/Parkin2 signaling pathway.

## 2. Materials and Methods

### 2.1. Reagent

BCP were purchased from Adamas dissolving in normal saline and 10% polyoxyethylated castor oil (EL) for preparing suspensions. For Western blotting and immunofluorescence staining, primary antibodies for LC3B, P62, Pink1, and TOM20 were purchased from Cell Signaling Technology (Beverly, NJ, USA), Parkin2 and β-actin for Proteintech Company (Wuhan, China), VDAC1 from Novus Biologicals (Centennial, CO, USA). Horseradish peroxidase (HRP)-conjugated antibodies (Proteintech Company, Wuhan, China) were used as secondary antibodies.

### 2.2. Cell Culture

The mouse neuron HT-22 cells were purchased from the Type Culture Collection of the Chinese Academy of Sciences (Shanghai, China). Dulbecco’s modified Eagle’s medium (DMEM)were purchased from Gibco. Fetal bovine serum (FBS) was purchased from Cell-Box. Penicillin and streptomycin were purchased from Genview. Cells were cultured in DMEM with 10% FBS, 1% penicillin and streptomycin. For incubation under normoxic condition, all the cells were cultured in a gas incubator at 37 °C with 5% CO_2_.

### 2.3. OGD/R Model

The different concentrations of BCP, including 0, 1, 5, 10, 15, 20 and 50 mM, were used to pre-treat HT22 cells (24 h, 37 °C), then the cells were subjected to OGD/R operation. Cells were seeded (1 × 10^6^ cells/well) with glucose-free medium, and cultured 2 h under hypoxic conditions (3% O_2_, 5% CO_2_, 92% N_2_). Then, the medium was changed into glucose-containing normal medium, and HT22 cells were cultured under normal conditions (95% air with 5% CO_2_, 37 °C). HT22 cells were analyzed after reoxygenation for 24 h.

### 2.4. MTT Assay

Cell viability was detected by MTT Cell Viability Assay kit (Beyotime Institute of Biotechnology). Cells were seeded into 96-well-plates with the density of 5 × 10^3^ cells in ever well and cultured with different concentrations of BCP (0, 1, 5, 10, 15, 20 and 50 mM) for 24 h at 37 °C. Cells were made the OGD/R operation prior to a 4 h incubation with 20 µL MTT/well. Then, DMSO was added to ever well to dissolve the formazan particles. These plates were rocked gently for 10 min and measured the absorbance at 490 nm with a microplate reader (Bio-RAD Model 550, Hercules, CA, USA).

### 2.5. Apoptosis Assay

HT-22 cells were seeded with the density of 1 × 10^6^ cells into ever well of 6-well plates and cultured for 24 h. Then, the preliminary treatment was the same as the above. Cells were collected and washed 3 times with cold PBS. The cells were resuspended in 100 μL of binding buffer containing 5 μL of Annexin V-FITC and 5 μL of PI for 10 min at room temperature in the dark. Next, the cells evaluated by flow cytometry in 488 nm.

### 2.6. Animals

Adult male mice C57BL/6 (20~25 g) were used in this study, which were obtained from the Experimental Animal Center (Chongqing Medical University), license number: SYXK Yu2018-0003 (2/22/2018). All protocols were under the permission of the Animal Experimental Committee in Chongqing Medical University.

### 2.7. Drug Treatment Procedures

The BCP suspensions and solvent were given orally according to mice’s body weight once daily for 5 days. The CIR mice model was operated after the final administration. The Sham group was injected with the same volume of saline, and mitophagy inhibitor Mdivi-1 (Selleck Chemicals, TX, USA) was injected intraperitoneally at the onset of reperfusion (3 mg/kg).

### 2.8. Middle Cerebral Artery Occlusion Model (MCAO/R Model)

Mice underwent procedures to cause transient focal cerebral ischemia via right MCAO. Briefly, mice were anaesthetized by 4% pentobarbital sodium (40 mg/kg, intraperitoneally). The tissue and blood vessels were bluntly separated and fixed with self-made small hooks. The right common carotid artery (CCA), external carotid artery (ECA) and internal carotid artery (ICA) were carefully separated and the proximal ends of the common and external carotid arteries were ligated and the internal carotid artery was clamped with an arterial clip. Inserting a 6-0 nylon monofilament (Jialing Biotechnology, Guangzhou, China) through the stump of the ICA into the ECA and advanced into the middle cerebral artery until feeling light resistance was felt (8~12 mm). At 1 h after occlusion, the nylon monofilament was withdrawn to realize reperfusion and the wounds were sutured, and then perfuse for 24 h. The mice of sham group underwent an identical procedure, but the nylon monofilament was not inserted. The mice were returned to clear cages and given free access to tap water and food at 37 ± 5 °C.

### 2.9. Evaluation of Neurological Function

After 24 h of reperfusion, the neurological function of the mice was scored as follows with reference to a modified Longa score. Zero points: no obvious neurological deficit; 1 point: when lifting the tail, the contralateral forelimb cannot extend; 2 points: spontaneous circling or walking to the contralateral side; 3 points: cannot bear the opposite side of its own weight, tripping to the damaged side; 4 points: no autonomous movement or disturbance of consciousness. The higher the score, the more severe the neurological impairment of the animal.

### 2.10. Rota-Rod Test

All mice were put on an accelerating rotarod cylinder. The speed of the rotary cylinder was increased from 4–40 rpm within 5 min. Blinded researcher recorded the time of staying on the rotarod cylinder for each mice. The trial should over when the mice fell off the rungs, gripped the rod, spun around for two revolutions without attempting to walk on the rod. Prior to MCAO operation, all mice were pre-trained for 3 days on a rotarod cylinder. The mean riding time of animals was recorded in thrice trials. Additionally, the ultimate average time on that day before the MCAO operation was considered the baseline. When reperfusion 24 h after MCAO operation, mice were put on the rotarod apparatus using the same methods.

### 2.11. Measurement of Infract Volume

2,3,5-Triphenyltetrazolium chloride (TTC)-stained brain sections were used to measure brain infarct volume. After 24 h of CIR, mice were anesthetized and executed. Brains were removed, placed at −20 °C for 20 min and cut into 4 coronal sections of 1 mm thicknesses. The samples were incubated in TTC (15 min, 37 °C), and fixed in 4% paraformaldehyde for overnight. Brain infarct volumes were calculated using Image J software, such as red as normal tissue and white as infarcted tissue.

### 2.12. H&E Staining

After 24 h of reperfusion, anesthetized mice were transcardially perfused with saline followed by 4% paraformaldehyde. The intact brain tissues were fixed for 24 hand then dehydrated in gradient concentrations of ethanol and xylene, then paraffin embedded, sliced into 5 µm thick sections and with H&E staining.

### 2.13. Real-Time PCR

The total RNA was isolated from MCAO/R mice hippocampus tissue using Total RNA Extractor (Trizol) (SangonBiotech, Shanghai, China). Total RNA was reverse-transcribed into cDNA using the MonScript™ RTIII All-in-One Mix with dsDNase (Monad, Suzhou, China). RNA concentrations were measured using a NanoDrop (Thermo Scientific, Waltham, MA, USA). Amplification was performed in an ABI Prism 7900HT system (Applied BioSciences, Waltham, MA, USA).The primer sequences were as follows: mouse mt-Atp6 (Fw: 5′-GCA GTC CGG CTT ACA GCT AA-3′; Rev: 5′-GGT AGC TGT TGG TGG GCT AA-3′) and mouse Rpl13 (Fw: 5′-GGA ACT AAA GCA GAC CCC GT-3′; Rev: 5′-CTG AGC CTA CAG CAG TGT CC-3′). Relative expression was presented as the mt-Atp6/Rpl13 ratio.

### 2.14. Immunofluorescence Staining

Slices were permeabilized in TBS + 0.03% Triton-X for 30 min, then blocked with goat serum for 1 h, and incubated with primary antibodies: LC3, Parkin2 and VDAC1 overnight at 4 °C. The primary antibody was recovered, a fluorescent secondary antibody was added, incubating for 1 h at room temperature. Finally, slices were stained with DAPI (Beyotime, Shanghai, China) for 10 min in a dark chamber at 37 °C, washed 3 times in PBS and cover slipped. The captured images were viewed with a fluorescence microscope (Olympus/BX51, Tokyo, Japan).

### 2.15. Western Blot

The hippocampus tissue lysates were prepared with RIPA buffer (Beyotime, Shanghai, China). For Western blot assay, the protein (40 μg/well) was separated in 10% SDS/PAGE gel and protein bands were transferred from gel onto a PVDF membrane (Millipore, Boston, MA, USA). The membranes were blocked for 2 h with 5% BSA, then incubated with primary antibodies: LC3, P62, Pink1, Parkin2, TOM20 and β-actin at 4 °C overnight. Next, the membranes were incubated with secondary antibodies for 2 h at 37 °C. Immunoreactive bands were detected using the enhanced chemiluminescence detection system (Bio-Rad, Hercules, CA, USA).

### 2.16. Transmission Electron Microscopy

After mice were anesthetized, PBS was injected into the left ventricle before fixation with 2.5% glutaraldehyde in paraformaldehyde. Fresh brain tissue was immediately removed and cut into ultrathin cortical sections, which were immersed in glutaraldehyde. Samples were glutaraldehyde with acetone and embedded by Epon812. Changes in neuronal ultrastructure were observed under TEM (Hitachi 7100, Tokyo, Japan).

### 2.17. Statistical Analysis

Results are shown as the means ± SD. SPSS v13 software (SPSS Inc., Chicago, IL, USA) was used for statistical analysis. Comparisons among multiple groups were performed with one-way ANOVA. A value of *p* < 0.05 was defined as statistically significant.

## 3. Results

### 3.1. BCP Alleviated OGD/R Injury-Induced Decreases of Cell Viability and Apoptosis

To examine the exact role of BCP on OGD/R injury, we first determined biologically safe doses of BCP for HT-22 cells under normoxic conditions using an MTT assay. The results showed that BCP concentrations ranging from 1 to 50 mM exerted no significant toxic effects on cell viability under normoxic conditions (Figure 1A). Subsequently, OGD/R-treated HT-22 cells were exposed to biologically safe concentrations of BCP (1, 5, 10, 15, 20, and 50 mM). We found that BCP obviously rescued cell viability following OGD/R injury in a dose-dependent manner, especially 10 mM BCP, which could significantly improve the decrease in cell viability caused by OGD/R injury relative to the control group (Figure 1B). Furthermore, the number and morphology of HT-22 cells after treatment with BCP at 5, 10, and 20 mM revealed alleviation of the effects of OGD/R injury (Figure 1C). In addition, apoptosis was reduced by treatment with BCP in a dose-dependent manner after OGD/R injury (Figure 1D,E). Collectively, these results indicate that BCP protected HT-22 cells against OGD/R injury.

### 3.2. Identification of Mitophagy Regulators by Metascape Online Analysis

Since activating mitophagy seems to be critical for neuronal survival following CIR injury, we selected 36 mitophagy regulators gene (Appendix A) from published papers to analyze their possible biological processes by Metascape online analysis (https://metascape.org/gp/index.html#/main/step1 (accessed on 22 April 2022). GO and KEGG enrichment analysis revealed these mitophagy regulators interrelated to biological processes such as pathways of neurogeneration, regulation of apoptotic process and growth which are closely connected with CIR-mediated neuroinjury (Figure 2A–D). Additionally, these essential mitophagy regulators interacted with multiple genes in PPI network (Figure 2E,F). We can found from Figure 2F that the PINK1-PARK pathway is close to the process of mitophagy. TOMMs and VDACs are the representative molecule of mitochondrion.BECN1, SQSTM1, ATG5, ATG12 and ULK1 are the representative molecule of autophagy. Therefore, we supposed that mitophagy may be the mechanism for BCP-mediated protection against CIR injury.

### 3.3. BCP Enhanced Mitophagy in OGD/R-Injured HT-22 Cells

To address the role of BCP in regulating mitophagy of OGD/R cells, the level of mitophagy was detected. First, we found that the OGD/R cells exhibited moderate decreases in protein expression of mitochondrial markers TOM20 and p62, and a moderate increase about the LC3-II/LC3-I ratio compared with the control cells. Notably, BCP treatment prominently decreased the expression of TOM20 and p62, and remarkably increased the LC3-II/LC3-I ratio compared with the OGD/R cells (Figure 3A–D). Furthermore, accumulation of autophagosomes was observed in OGD/R cells compared with the control cells. However, numbers of autophagosomes in cells treated with BCP were significantly increased, and autophagic vacuoles wrapping around damaged mitochondria and organelles could be seen that were absent in the OGD/R cells (Figure 3E,F). In addition, double-immunofluorescence staining of LC3 and the VDAC1 (a mitochondrial marker) was investigate to colocalization of mitophagy. We observed less fluorescence of LC3 in the control group. After the OGD/R operation, increased LC3 expression suggested a higher level of autophagy. Moreover, the expression of LC3B/VDAC1 were higher in the OGD/R cells compared with the control cells, suggesting that mitophagy was activated. Significantly higher expression of LC3B/VDAC1 were observed in the BCP treatment cells compared with the OGD/R cells (Figure 3G,H). Quantitative analysis also suggested increased the expression of LC3B/VDAC1 in the BCP treatment cells compared with the other cells. Finally, qRT-PCR revealed a moderate decrease of mtDNA copy numbers in the OGD/R cells compared with the control cells, and mtDNA copy numbers were sharply diminished in the BCP treatment cells compared with the OGD/R cells (Figure 3I). These results illustrated that BCP pre-treatment was important for facilitating mitophagy to clear damaged mitochondria in OGD/R cells.

### 3.4. BCP Activates the Pink1/Parkin2 Pathway in OGD/R-Injured HT-22 Cells

To examine the mechanism by which BCP mediated mitophagy in OGD/R cells, effects of BCP on the Pink1/Parkin2 pathway were observed. Our results show that the OGD/R group exhibited moderate increases in Pink1 and Parkin2 protein expression levels compared with the control group, while BCP treatment significantly increased Pink1 and Parkin2 protein expression levels compared with OGD/R and control groups (Figure 4A–C).

Parkin2 is recruited to mitochondria, where is remains localized during mitophagy. Double-immunofluorescence staining results shown that Parkin2 was expressed at low levels during mitochondrial translocation in the control group, while moderate increases the expression of Parkin2/VDAC1 were observed in the OGD/R cells. Additionally, significant increases the expression of Parkin2/VDAC1 were observed in the BCP-treated OGD/R cells (Figure 4D,E). These results implicated that the Pink1/Parkin2 signaling pathway is activated in OGD/R cells.

### 3.5. Facilitating Mitophagy via the Pink1/Parkin2 Pathway Plays an Important Role in BCP Protection against OGD/R Injury of HT-22 Cells

To clarify the involvement of activated mitophagy in BCP-mediated protection against OGD/R injury of HT-22 cells, mitophagy was blocked with the mitophagy inhibitor Mdivi-1. MTT assay results showed that BCP treatment markedly promoted the growth and survival of OGD/R cells, whereas Mdivi-1 obviously blocked BCP-induced growth and survival of OGD/R cells (Figure 5A). Moreover, BCP treatment alleviated the number and morphology of OGD/R cells, whereas Mdivi-1 blocked the protective role of BCP against these changes in OGD/R cells (Figure 5B). Furthermore, BCP treatment reduced apoptosis of OGD/R cells, a feature that was also blocked by Mdivi-1 (Figure 5C,D).

Additionally, we evaluated mitophagy levels after exposure to Mdivi-1. We found that Mdivi-1 reversed the BCP-mediated decreases in expression of TOM20 and p62, and increase in the LC3-II/LC3-I protein ratio in OGD/R cells (Figure 6A–D). Consistently, Mdivi-1 blocked the BCP-induced increase of autophagic vacuoles wrapping around damaged mitochondria in OGD/R cells (Figure 6E,F). Furthermore, Mdivi-1 blocked the significant BCP-induced increase of Parkin2/VDAC1-positive cells following OGD/R injury (Figure 6G,H). In addition, Mdivi-1 reversed the BCP-mediated decrease in mtDNA copy numbers in OGD/R cells (Figure 6I). Finally, Mdivi-1 reversed BCP-mediated increases in Pink1 and Parkin2 protein expression levels in OGD/R cells (Figure 7A–C). In accordance with mitochondrial translocation assay results, Mdivi-1 blocked BCP-mediated increases of Parkin2/VDAC1-positive cells (Figure 7D,E). Therefore, BCP mediated protection against OGD/R-mediated injury in HT-22 cells mainly by facilitating mitophagy via the Pink1/Parkin2 pathway.

### 3.6. BCP Protected against Cerebral Ischemia-Reperfusion Injury by Facilitating Mitophagy in C57BL/6 Mice

To determine whether BCP contributes to protection against CIR in a manner dependent on mitophagy in vivo, as shown in Figure 8A, C57BL/6 mice were divided into groups treated with 36 mg/kg, 72 mg/kg, or 144 mg/kg BCP (low-, middle-, and high-dose groups) before undergoing a transient MCAO procedure. Mdivi-1 was injected intraperitoneally at the onset of reperfusion (3 mg/kg). Mice were sacrificed 24 h after reperfusion. Severe neurological deficits were present in the MCAO/R group compared with sham and BCP-treated sham groups, and BCP treatment reduced neurological scores in a dose-dependent manner (Figure 8B). MCAO/R group mice exhibited reduced times in rotarod testing compared with respective sham and BCP-treated sham groups. Treatment with BCP increased times spent on the rotarod after MCAO/R in a dose-dependent manner (Figure 8C), especially 72 mg/kg BCP, which prominently protected against MCAO/R-mediated nerve injury. However, Mdivi-1 abolished these neuroprotective effects, as indicated by increased neurologic scores and reduced times on the rotarod (Figure 8B,C). Infarct volumes were assessed by TTC. Compared with outcomes in the MCAO/R group, BCP treatment markedly decreased infarct volumes in a dose-dependent manner, suggesting that BCP improved functional outcomes after MCAO/R injury (Figure 8D,E). In addition, H&E stained sections were analyzed the necrotic cells in the hippocampal CA1 region. As shown in Figure 8F, neurons in the hippocampal CA1 region were arranged in nomal rows, shown intact cell structures, and had clear nuclear outlines in sham and BCP-treated sham groups (black arrows). After 24 h of reperfusion, neurons were necrotic and shrank, cellular edema, and nuclear condensation (red arrows). However, BCP significantly prevented neuron damage and improved the pathology associated with MCAO/R in a dose-dependent manner. Interestingly, Mdivi-1 blocked this neuroprotection in MCAO/R mice, as indicated by increased infarct volumes and CA1 neuronal injury (Figure 8D–F). These results suggest that BCP protected against MCAO/R injury in vivo in an autophagy-dependent manner.

### 3.7. BCP Facilitated Mitophagy after Cerebral Ischemia-Reperfusion Injury in C57BL/6 Mice

Western blot results showed moderate decreases in protein expression of mitochondria markers TOM20 and p62, as well a moderate increase in the LC3-II/LC3-I protein ratio in the MCAO/R group compared with the sham group. Compared with the MCAO/R group, BCP pretreatment significantly increased the LC3-II/LC3-I protein ratio and TOM20 expression, and decreased p62 expression. However, changes in expression of these proteins were reversed by treatment with a mitophagy inhibitor (Mdivi-1) (Figure 9A–D). In addition, electron microscopy results showed that healthy mitochondria were present in the sham group. Neurons in the MCAO/R group displayed mitochondrial swelling, loss of matrix density, autophagic vacuoles, partially degraded mitochondria, small numbers of autophagic vacuoles, and double-membraned autophagosomes containing damaged mitochondria. In contrast, mitochondria with some slight swelling and nearly normal matrix densities were observed in the BCP group. Additionally, typical autophagosomes were frequently observed in the BCP group, and were more abundant compared with the MCAO/R group. In contrast, only a small number of autophagosomes were found in the BCP + Mdivi-1 group and no typical autophagosomes were observed in the Mdivi-1 group (Figure 9E,F).

We further detected expression of the mitochondrial membrane protein VDAC1 and LC3 by immunofluorescence co-staining. As reflected in Figure 9G,H, there was little increase in LC3/VDAC1-positive cells in the control group, while BCP pretreatment significantly increased the percentage of LC3/VDAC1-positive cells compared with the MCAO/R group. In contrast, the percentage of LC3/VDAC1-positive cells was clearly decreased in the BCP + Mdivi-1 group. Finally, we examined relative amounts of mitochondria in different groups by qPCR. mt-Atp6/Rpl13 ratios were significantly decreased in the BCP group compared with the MCAO/R group, confirming that the loss of mitochondria was due to autophagy. Employment of the mitophagy inhibitor Mdivi-1 significantly increased the mt-Atp6/Rpl13 ratio (Figure 9I). Collectively, these results revealed that BCP can clear MCAO/R-induced damaged mitochondria by accelerating mitophagy.

### 3.8. Pink1/Parkin2 Pathway-Dependent Mitophagy Is Implicated in BCP-Induced Protection against CIR Injury

We detected expression of Pink1 and Parkin2 proteins by Western blot. The results suggested an increase in Pink1 and Parkin2 protein expression levels in the MCAO/R group compared with the sham group. The BCP treatment group exhibited significantly upregulated expression of Pink1 and Parkin2 compared with the MCAO/R group (Figure 10A–C). However, administration of Mdivi-1 inhibited BCP-induced upregulation of Pink1 and Parkin2 expression levels. In addition, we examined the recruitment of Parkin2 to mitochondria. Parkin2 and VDAC1 localization were detected by immunofluorescence staining in ischemic hippocampi of MCAO/R mice. We found that translocation of Parkin2 to mitochondria was increased in the BCP group compared with the MCAO/R group (significant increase of Parkin2/VDAC1-positive cells). In the BCP + Mdivi-1 group, translocation of Parkin2 to mitochondria was decreased compared with findings from the BCP group (Figure 10D,E). Therefore, these results indicated that the Pink1/Parkin2 signaling pathway is required for BCP-induced mitophagy activation in BCP-induced protection against CIR injury.

## 4. Discussion

CIR injury is an extremely complex pathophysiological process comprising a rapid cascade of intracellular calcium overload, lipid peroxidation, oxygen free radical damage, apoptotic gene activation, excitotoxicity (excitatory amino acids-induced neurotoxicity), and inflammatory cytokine damage [25,26]. The involved mechanisms, such as mitochondrial dysfunction, inflammatory injury, and cell damage, play different roles in the process of CIR injury. Previous studies showed that mitochondrial damage and dysfunction during CIR are important causes of neuronal cell death [27]. Current studies suggest that a large amount of cytochrome C is released from mitochondria into cells by depolarization of the mitochondrial membrane and opening of the mitochondrial permeability transition pore in CIR injury, leading to mitochondrial dysfunction [28,29]. Further oxidative damage of mitochondria is induced by ATP synthesis deficiency and new reactive oxygen species in CIR injury [30]. Therefore, timely clearance of damaged mitochondria is essential to mitigate CIR-induced brain damage. Mitophagy is a type of selective autophagy that can maintain homeostasis of the intracellular environment by specifically recognizing and removing abnormal mitochondria [5]. Although many studies have investigated the effect of mitochondrial autophagy in CIR injury, the exact role mitochondrial autophagy plays in this process is unknown. It remains controversial whether mitophagy can play a protective role in ischemic preconditioning and/or exert different effects on how reperfusion injury occurs [24,31]. A recent study suggests that carnosine can improve mitochondrial function and reduce CIR injury by downregulating mitophagy [32]. Another study showed that upregulating mitophagy can protect against CIR injury by promptly removing damaged mitochondria [33]. Accumulating research indicates that mitophagy are activated in a stress-induced pathway to clear damaged mitochondria in CIR injury. Furthermore, mitophagy aggravates brain injury during the ischemia phase, but alleviates brain injury in the reperfusion phase. In addition, the protective role of mitophagy during reperfusion may be attributable to mitochondrial clearance and inhibition of downstream apoptosis. Accordingly, insufficient clearance of damaged mitochondria causes cell death [34]. In this study, we found that the CB2R agonist BCP, a natural sesquiterpene compound, protected against CIR injury both in vitro and in vivo in a dose-dependent manner. Consistent with activation of mitophagy after CIR injury, BCP treatment significantly upregulated CIR-induced mitophagy, yielding a neuroprotective effect after CIR injury. In addition, the mitophagy inhibitor Mdivi-1 successfully blocked BCP-induced protection against CIR injury in both OGD/R in vitro and MCAO/R in vivo models. Therefore, mitophagy may be an important target for the treatment of CIR injury.

LC3-I is converted to lipidated LC3-II, a classic hallmark of autophagy [35]. p62, a selective substrate protein degraded by autophagy, binds to Atg8/LC3 on autophagy membranes to target autophagosomes for degradation during autophagy. Therefore, ubiquitin-p62-LC3 can perform autophagic degradation on damaged phagocytes and eliminate defective membranous organelles and proteins [36]. TOM20 is a mitochondrial outer membrane protein and subunit of the important TOM complex receptor. The level of TOM20 reflects the relative number of mitochondria. Hence, relative expression of TOM20 and p62, as well as the ratio of LC3-II/LC3-I proteins, are used to evaluate levels of mitophagy [35]. In this study, when BCP treatment was given, the LC3-II/LC3-I protein ratio sharply increased while expression of p62 and TOM20 were significantly decreased in both OGD/R in vitro and MCAO/R in vivo models of CIR injury. These results indicate that mitochondrial autophagy levels were further increased. The process of autophagy/mitophagy occurs sequentially as follows: phagophore formation, fusion of the blind ends of phagophores, autophagosome formation, fusion of autophagosomes with lysosomes, and transformation of autophagosomes into autolysosomes; during the final step of autophagy, residual bodies can be detected [37]. During this process, various damaged proteins or organelles, including mitochondria, are wrapped by double-layered membrane autophagy vesicles that are then sent to lysosomes (animals) or vacuoles (yeast and plants), which are eventually degraded and recycled [38]. Autophagosomes are characterized by a double-layer or multilayer membrane-like vacuole structure that contains cytoplasmic components, such as mitochondria, and can be observed by electron microscopy [39]. In this study, TEM results revealed the presence of autophagosomes with a typical bilayer membrane structure in the BCP treatment group in both OGD/R in vitro and MCAO/R in vivo models of CIR injury.

Pink1-Parkin2-mediated mitophagy is a classic pathway to clear damaged mitochondria [40]. Release of Pink1 from the mitochondrial outer membrane to the inner membrane is blocked by mitochondrial membrane potential depolarization, meaning Pink1 cannot be degraded and instead accumulates on the outer membrane of mitochondria. When Pink1 aggregates on the mitochondrial outer membrane, various proteins including Parkin, ubiquitin, and TBK1 (TANK-binding kinase 1) are recruited and activated. Ample studies have investigated the effect of Pink1/Parkin2-mediated mitophagy in CIR injury. Previous reports suggest that mitophagy is activated in CIR-induced brain damage via the PINK1/PARK2/p62 signaling pathway [41]. Another study showed that electroacupuncture ameliorates neuronal injury by Pink1/Parkin-mediated mitophagy clearance in CIR [17]. In the present experiment, we demonstrated that BCP pretreatment can significantly increase expression of Pink1 and Parkin2, suggesting that BCP-induced mitochondrial autophagy is dependent on the Pink1/Parkin2 pathway. Taken together, these results suggest that BCP reduced CIR injury in both OGD/R in vitro and MCAO/R in vivo models by upregulating CIR-induced mitophagy through effects on the Pink1/Parkin2 signaling pathway.

## 5. Conclusions

Our study indicates that mitophagy was activated following CIR injury and BCP pretreatment could further activate CIR-induced mitophagy to clear damaged mitochondria to reduce brain damage. This protective mechanism against CIR injury (Figure 11) occurs through recruitment of Pink1 and Parkin2 on the outer membrane of mitochondria, which increased the LC3-II/LC3-I protein ratio and decreased expression of p62. Subsequently, formation of phagophores containing damaged mitochondria occurs, which fuse with lysosomes and transform into autolysosomes. By this mechanism, damaged mitochondria can be degraded to produce amino acids and other nutrients for neurons.

## Figures and Tables

**Figure 1 brainsci-12-00868-f001:**
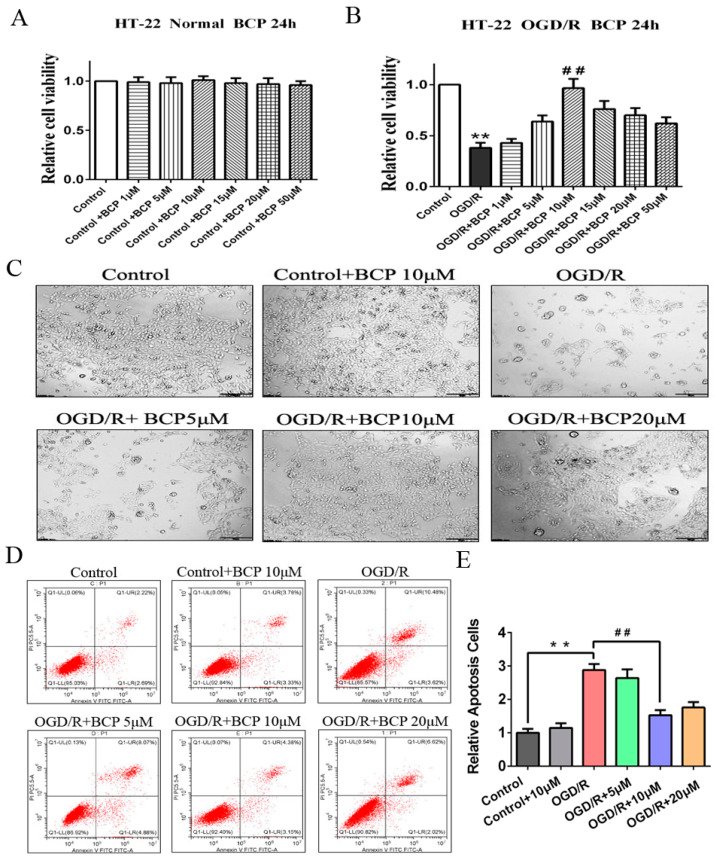
BCP alleviated OGD/R injury-induced decreases of cell viability and apoptosis. (**A**) Role of BCP on cell viability of HT-22 cells (normal condition). (**B**) Role of BCP on cell viability of OGD/R cells. (**C**) HT-22 cells were observed by microscopy after treatment with varying doses of BCP (OGD/R, 24 h). (**D**) BCP reduced the apoptosis of OGD/R cells, as indicated by flow cytometry. (**E**) Quantitative analysis of flow cytometry results. (*n* = 3; ** *p* < 0.01 OGD/R group vs. control group; ## *p* < 0.01 OGD/R + BCP 10 μM group vs. OGD/R group).

**Figure 2 brainsci-12-00868-f002:**
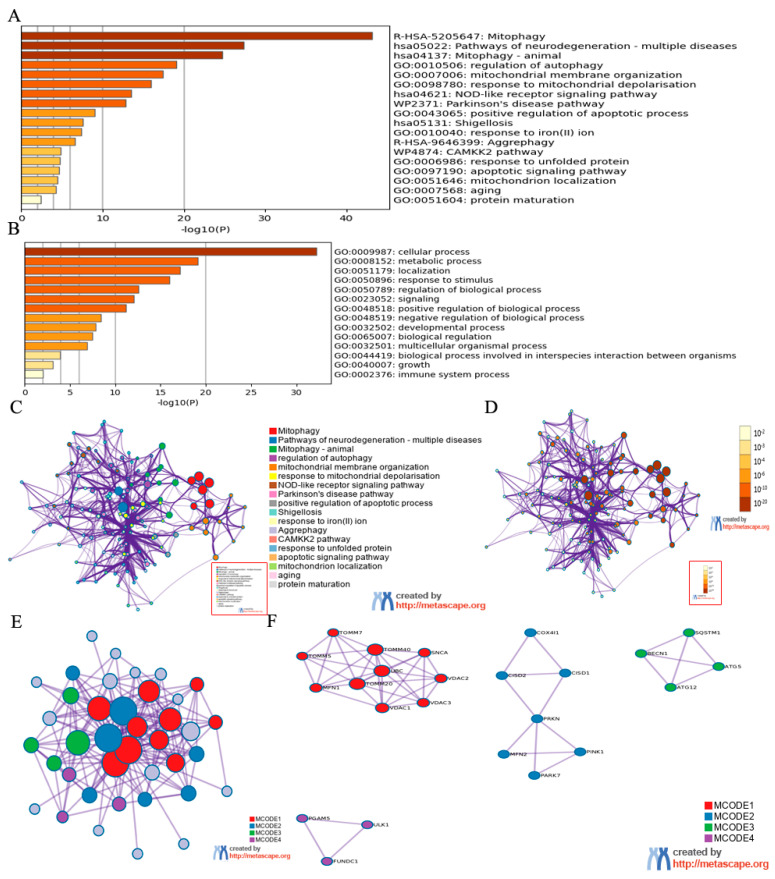
Identification of mitophagy regulators by Metascape Online Analysis. (**A**) Bar graph of enriched terms across input mitophagy regulator lists, colored by *p*-values. (**B**) The top-level Gene Ontology biological processes can be viewed here. (**C**) Network of enriched terms: colored by cluster ID, where nodes that share the same cluster ID are typically close to each other. (**D**) Colored by *p*-value, where terms containing more genes tend to have a more significant *p*-value. (**E**,**F**) Protein-protein interaction network and MCODE components identified in the mitophagy regulator lists.

**Figure 3 brainsci-12-00868-f003:**
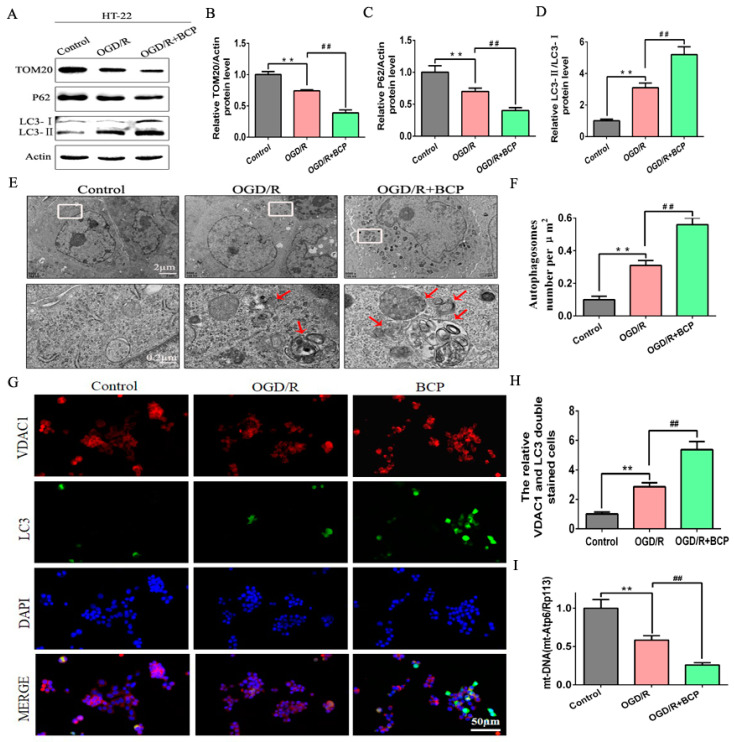
BCP enhanced mitophagy in OGD/R-injured HT-22 cells. (**A**–**D**) Representative Western blots showing protein levels of p62 and TOM20, and the protein ratio of LC3-II/LC3-I. p62 and TOM20 protein levels were normalized to β-actin. (**E**,**F**) Transmission electron microscopy revealed that BCP pretreatment increased numbers of autophagosomes in HT-22 cells after OGD/R. Red arrows indicate autophagosomes. (**G**,**H**) Double-immunofluorescence staining for LC3B and VDAC1 in HT-22 cells at 24 h after reperfusion, VDAC1 (red) and LC3B (green) double-stained cells (yellow) indicate mitophagy (scale bar = 50 μm, 400× magnification). (**I**) Relative mitochondrial DNA levels indicated by the ratio of mt-Atp6 were assessed by qRT-PCR. (*n* = 3; ** *p* < 0.01 OGD/R group vs. control group; ## *p* < 0.01 OGD/R + BCP group vs. OGD/R group).

**Figure 4 brainsci-12-00868-f004:**
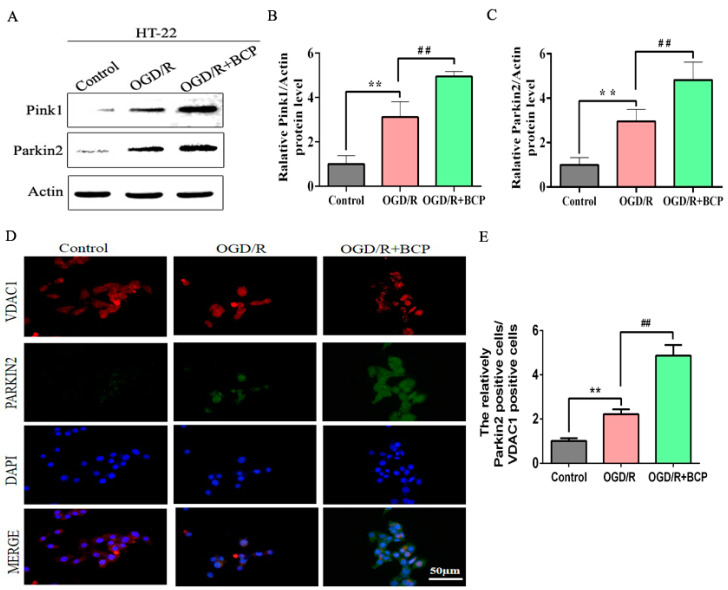
BCP activates the Pink1/Parkin2 pathway in OGD/R-injured HT-22 cells. (**A**–**C**) Representative Western blots showing protein levels of Pink1 and Parkin2. Protein levels were normalized to β-actin. (**D**,**E**) Double-immunofluorescence staining for VDAC1 and Parkin2 in HT-22 cells at 24 h after reperfusion. VDAC1 (red) and Parkin2 (green) double-stained cells (yellow) indicate mitophagy. (Scale bar = 50 μm, 400× magnification). (*n* = 3; ** *p* < 0.01 OGD/R group vs. control group; ## *p* < 0.01 OGD/R + BCP group vs. OGD/R group).

**Figure 5 brainsci-12-00868-f005:**
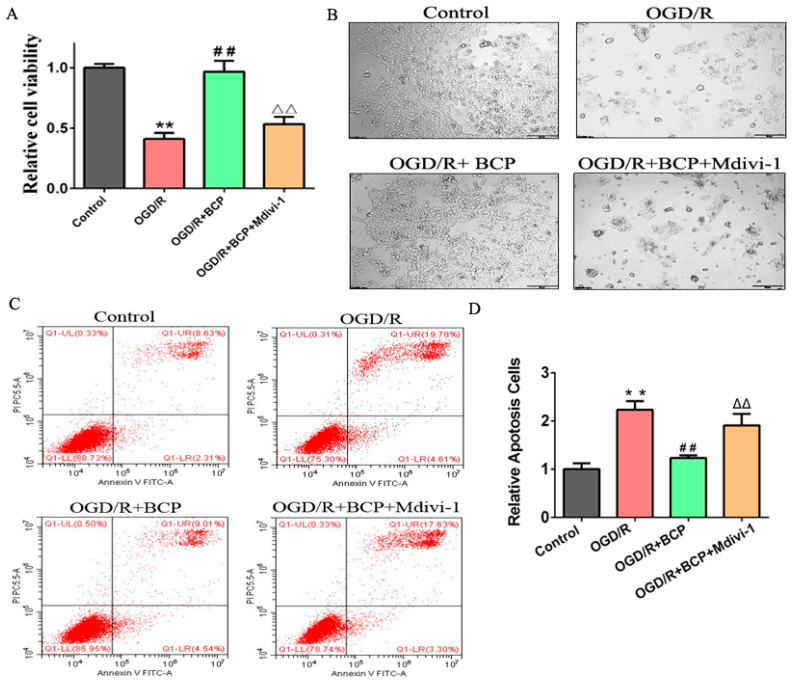
Facilitating mitophagy plays an important role in BCP protection against OGD/R injury of HT-22 cells. (**A**) Effect of BCP on cell viability of OGD/R-injured HT-22 cells after treatment with the mitophagy inhibitor Mdivi-1. (**B**) HT-22 cells were observed by microscopy after treatment with Mdivi-1. (**C**,**D**) Flow cytometry and quantitative analysis of apoptosis of OGD/R cells after treatment with Mdivi-1. (*n* = 3; ** *p* < 0.01 OGD/R group vs. control group; ## *p* < 0.01 OGD/R + BCP group vs. OGD/R group; △△ *p* < 0.01 Mdivi-1 group vs. BCP group).

**Figure 6 brainsci-12-00868-f006:**
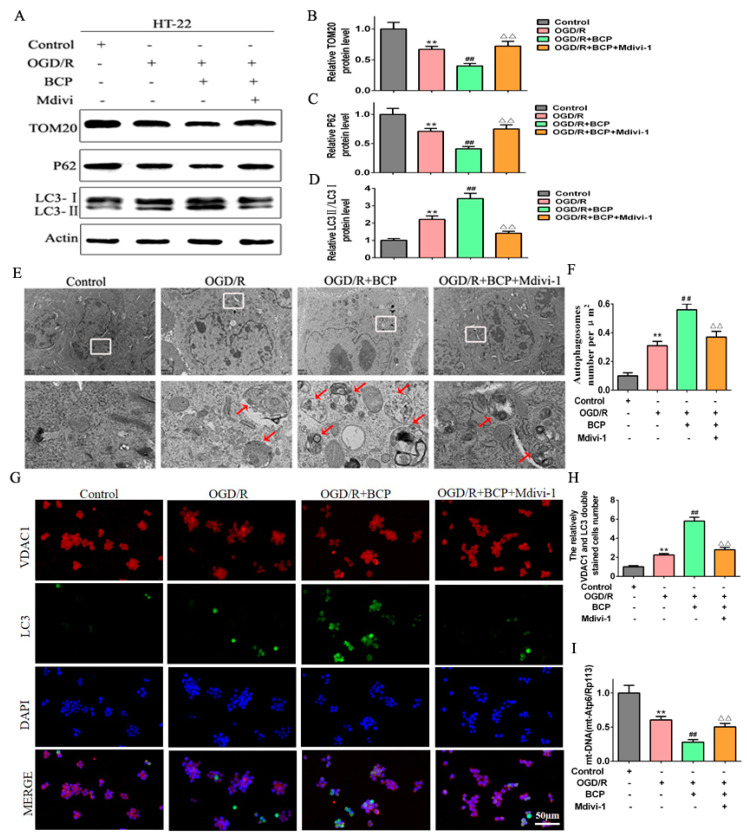
BCP-mediated mitophagy was suppressed by a mitophagy inhibitor. (**A**–**D**) Representative Western blots showing protein levels of p62 and TOM20, and the protein ratio of LC3-II/LC3-I after treatment with the mitophagy inhibitor Mdivi-1. p62 and TOM20 protein levels were normalized to β-actin. (**E**,**F**) Transmission electron microscopy revealed that BCP pretreatment increased numbers of autophagosomes in HT-22 cells after treatment with Mdivi-1. (Scale bar = 2 μm). Red arrows indicate autophagosomes. (**G**,**H**) Double immunofluorescence staining for LC3B and VDAC1 in HT-22 cells after treatment with Mdivi-1. VDAC1 (red) and LC3B (green) double-stained cells (yellow) indicate mitophagy (scale bar = 50 μm, 400× magnification). (**I**) Relative mitochondrial DNA levels indicated by the ratio of mt-Atp6/Rpl13 were assessed by qRT-PCR. (*n* = 3; ** *p* < 0.01 OGD/R group vs. control group; ## *p* < 0.01 OGD/R + BCP group vs. OGD/R group; △△ *p* < 0.01 Mdivi-1 group vs. BCP group).

**Figure 7 brainsci-12-00868-f007:**
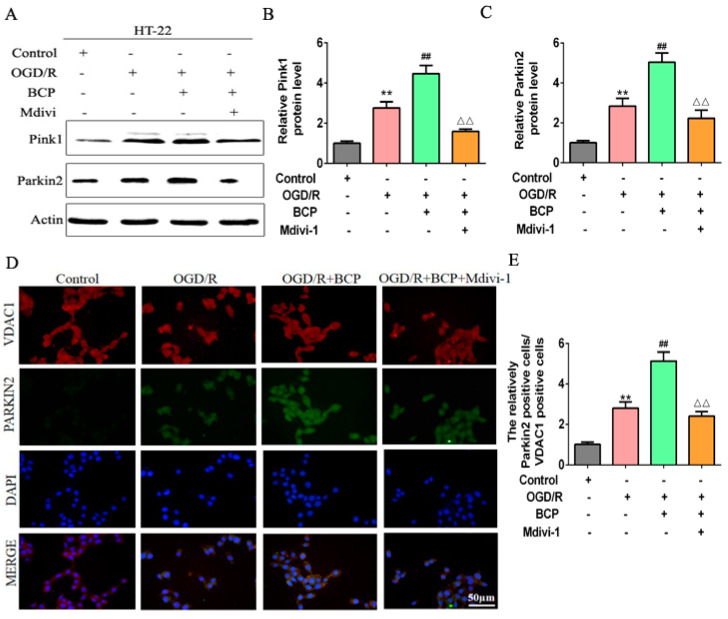
Pink1/Parkin2 pathway-dependent mitophagy is implicated in BCP-induced mitophagy. (**A**–**C**) Representative Western blots showing protein levels of Pink1 and Parkin2 after treatment with the mitophagy inhibitor Mdivi-1. Protein levels were normalized to β-actin. (**D**,**E**) Double-immunofluorescence staining for VDAC1 and Parkin2 in HT-22 cells after treatment with Mdivi-1. VDAC1 (red) and Parkin2 (green) double-stained cells (yellow) indicate mitophagy. (Scale bar = 50 μm, 400× magnification). (*n* = 3; ** *p* < 0.01 OGD/R group vs. control group; ## *p* < 0.01 OGD/R + BCP group vs. OGD/R group; △△ *p* < 0.01 Mdivi-1 group vs. BCP group).

**Figure 8 brainsci-12-00868-f008:**
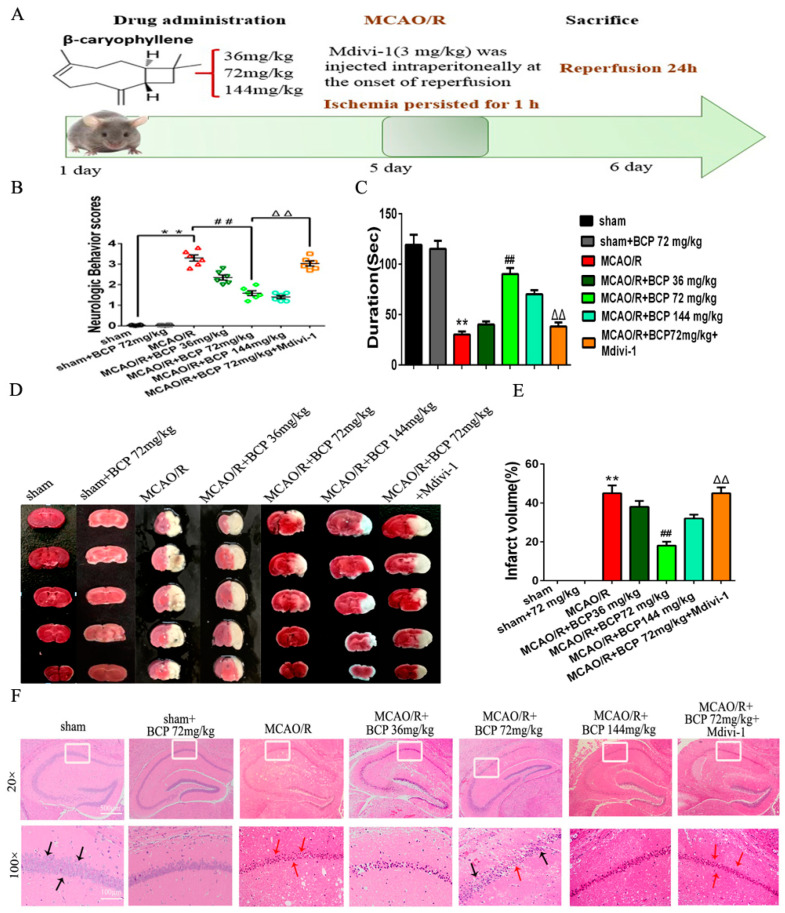
BCP protected against cerebral ischemia-reperfusion injury by facilitating mitophagy in C57BL/6 mice. (**A**) Schematic overview of experimental process. (**B**) Quantitative analysis of neurological function. (**C**) Duration of the rotarod test for all mice. (**D**) TTC staining of representative sections. (**E**) Quantification of infarcted volume. (**F**) H&E images of representative hippocampal CA1 sections of each group. ((**A**–**F**), *n* = 6; ** *p* < 0.01 MCAO/R group vs. sham group; ## *p* < 0.01 MCAO/R + BCP group vs. MCAO/R group; △△ *p* < 0.01 Mdivi-1 group vs. BCP group).

**Figure 9 brainsci-12-00868-f009:**
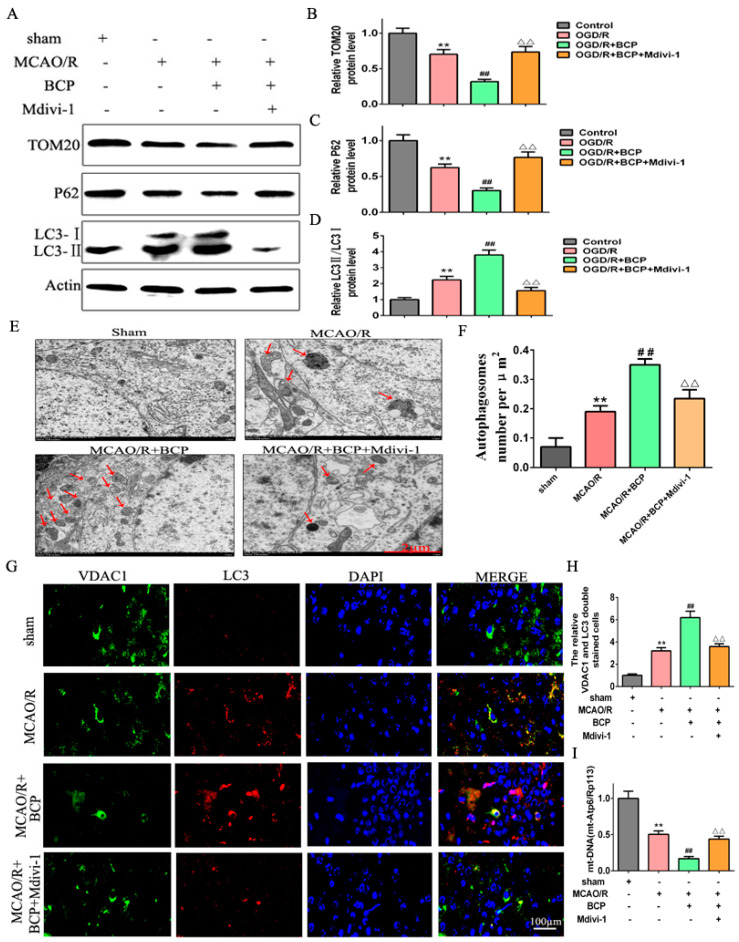
BCP facilitated mitophagy after CIR injury in C57BL/6 mice. (**A**–**D**) Representative Western blots showing protein levels of p62 and TOM20, and the protein ratio of LC3-II/LC3-I. p62 and TOM20 protein levels were normalized to β-actin. (**E**,**F**) Transmission electron microscopy revealed numbers of autophagosomes in hippocampal CA1 sections of MCAO/R mice. Red arrows indicate autophagosomes. (**G**,**H**) Double-immunofluorescence staining for LC3B and VDAC1 in ischemic hippocampal CA1 sections after reperfusion. VDAC1 (red) and LC3B (green) double-stained cells (yellow) indicate mitophagy (scale bar = 100 μm, 400× magnification). (**I**) Relative mitochondrial DNA levels (indicated by the ratio of mt-Atp6 to Rpl13) were assessed by real-time PCR. ((**A**–**I**), *n* = 6; ** *p* < 0.01 MCAO/R group vs. sham group; ## *p* < 0.01 MCAO/R + BCP group vs. MCAO/R group; △△ *p* < 0.01 Mdivi-1 group vs. BCP group).

**Figure 10 brainsci-12-00868-f010:**
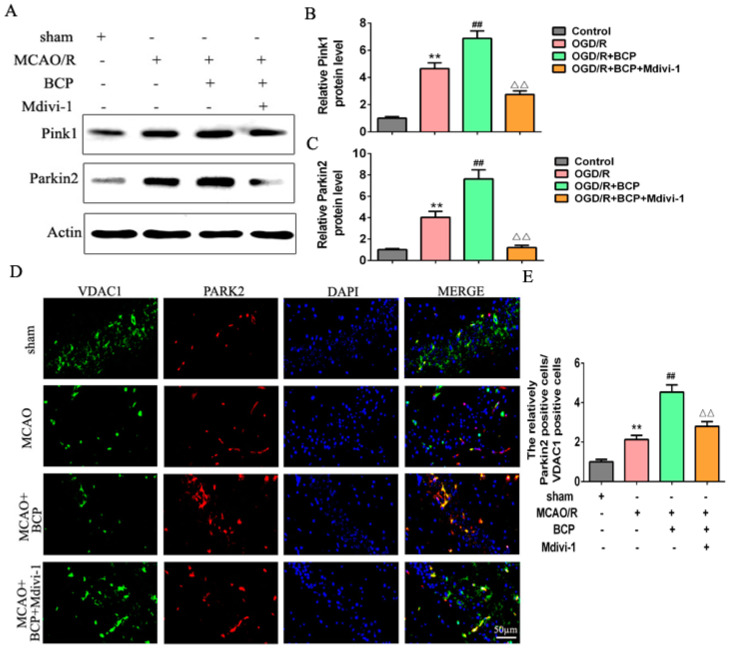
Pink1/Parkin2 pathway-dependent mitophagy is implicated in BCP-induced protection against CIR injury. (**A**–**C**) Representative Western blots showing protein levels of Pink1 and Parkin2. Protein levels were normalized to β-actin. (**D**,**E**) Double-immunofluorescence staining for VDAC1 and Parkin2 in the penumbra of ischemic hippocampal CA1 sections at 24 h after reperfusion. VDAC1 (red) and Parkin2 (green) double-stained cells (yellow) indicated mitophagy. (Scale bar = 50 μm, 400× magnification). ((**A**–**E**), *n* = 6, ** *p* < 0.01 MCAO/R group vs. sham group; ## *p* < 0.01 MCAO/R + BCP group vs. MCAO/R group; △△ *p* < 0.01 Mdivi-1 group vs. BCP group).

**Figure 11 brainsci-12-00868-f011:**
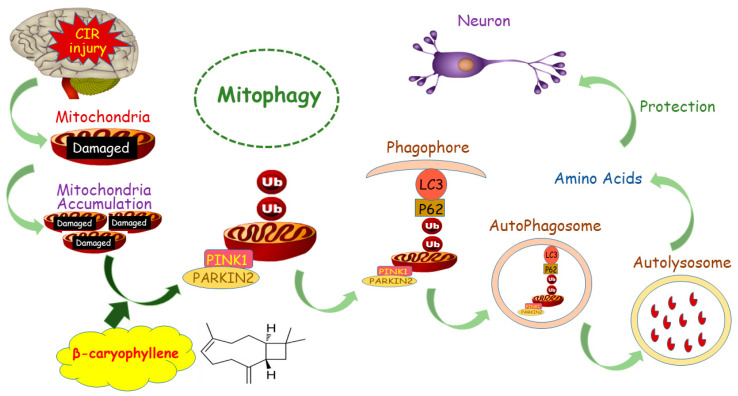
Molecular mechanism of the neuroprotective effect of BCP against CIR-induced neuronal injury. BCP pretreatment could further activate CIR-induced mitophagy to clear damaged mitochondria to reduce brain damage in CIR injury by recruiting Pink1 and Parkin2 on the outer membrane of mitochondria.

## Data Availability

All the data are contained within the article.

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
