# Peer review of "Facilitating Mitophagy via Pink1/Parkin2 Signaling Is Essential for the Neuroprotective Effect of β-Caryophyllene against CIR-Induced Neuronal Injury"

_brainsci, 2022, doi:10.3390/brainsci12070868_

Round 1
Reviewer 1 Report
This paper explores the role of mitophagy via Pink1/Parkin2 signalling in the protective effect of beta-caryophyllene against cerebral ischemia-reperfusion injury. The paper is well presented but requires clarification of some issues:
In the methods, group numbers for the HE staining is presented, what were the numbers used for the other experiments?
Was a multiple-comparison correction used for post-hoc tests following the ANOVA, such as a Bonferroni correction?
In Figure 2, C through F are not able to be interpreted due to the poor image quality and small figure legend.
The final paragraph of the Discussion is repeated for the conclusion.
Some sentences require revision for grammar:
- Line 278
- Line 297
- Pink1 does not require the full name on line 657
Author Response
Please see the attachment.
First, we would like to thank the editorial board member and the expert external reviewers for the constructive comments on the paper entitled “Facilitating Mitophagy via Pink1/Parkin2 Signaling is Essential for the Neuroprotective Effect of β-caryophyllene against CIR-Induced Neuronal Injury”( brainsci-1779005). We have carefully addressed the critiques in a point‐to‐point manner as ascribed below as well as in the revised manuscript. Changes made to the manuscript are underlined.

Reviewer 2 Report
This paper discusses the neuroprotective effect of b-caryophyllene in CIR-induced neuronal injury via the Pink1/Parkin2 mitophagy signaling pathway. This manuscript is well written, and the data align with the aims and objectives of this study. I have read the manuscript and have a few minor suggestions with regards to this study.
Comments:
1) Histology figures: Would it be possible for the authors to submit a colored image of figures? There is not much contrast in any of the histology figures and therefore it is hard to discern the viability of cells in the control, IR affected and treatment groups. Also, the quality of images is not the best for histology and therefore clearer images would be more appreciated.
2) Fig 2C-F: These network data will have to be completely rescaled and better or high-quality images will have to be provided. The images are not clear at all, and it was impossible to see what genes or pathways are connected to each other. Since this image is vital to the study better quality images will be needed.
3) Fig 7.D: While the authors have shown that Parkin levels increase during IR and Mdivi decreases its levels, this panel does not conclusively reflect that opinion. The Parkin levels in OGD/R, OGD/r-BCP, and OGD/R-BCP-Mdivi look almost the same. Do the authors have other fields of view that show increase or decrease of Parkin levels that reflect their other data presented in their paper?
4) Fig. 8D: From this figure it appears that BCP at 72 mg/kg seems to be more effective than that twice that dose and the infarct size with a dose of 144 mg/kg or the same dose with Mdivi look almost the same. Do the authors have data with a dose of 72 mg/kg treated with Mdivi? That data would give a better perspective of the dose response effects and the Mdivi treatment.
5) Fig. 8F: Could the authors show an H&E image of BCP 144 mg/kg treated with Mdivi? Moreover, treatment of MCAO/R mice with BCP+Mdivi at 72 mg/kg seems very similar to treatment at 144 mg/kg and it does not seem to reverse the beneficial effect to a great extent. Could the authors comment more on this observation? Also, it would greatly benefit from adding arrows indicating edema or other pathophysiology for better understanding for the reader.
6) Fig 9E: Could the authors point arrows to indicate mitochondria in the EM structures? The EM images are not high quality and therefore difficult to compare treated with sham.
7) Fig. 9G: The levels of LC3 shows progressive increase and decrease with BCP treatment and BCP+Mdivi, respectively. However, the quantification for LC3 shows that the levels decreased by almost half when treated with Mdivi. However, the image shows even lesser quantity of LC3 than what is reported in the quantification. Could the authors comment on the method of quantification?
Author Response

(The authors gave the same response as above.)
